# RNA compaction and iterative scanning for small RNA targets by the Hfq chaperone

Ewelina M. Małecka [1,2] ✉ & Sarah A. Woodson [1] ✉

RNA-guided enzymes must quickly search a vast sequence space for their targets. This search is aided by chaperones such as Hfq, a protein that mediates regulation by bacterial small RNAs (sRNAs). How RNA binding proteins enhance this search is little known. Using single-molecule Förster resonance energy transfer, we show that *E. coli* Hfq performs a one-dimensional scan in which compaction of the target RNA delivers sRNAs to sites distant from the location of Hfq recruitment. We also show that Hfq can transfer an sRNA between different target sites in a single mRNA, favoring the most stable duplex. We propose that compaction and segmental transfer, combined with repeated cycles of base pairing, enable the kinetic selection of optimal sRNA targets. Finally, we show that RNA compaction and sRNA transfer require conserved arginine patches. We suggest that arginine patches are a widespread strategy for enabling the movement of RNA across protein surfaces.

RNA-guided targeting of nucleic acids is a versatile tool for gene regulation and viral defense in all kingdoms of life. In bacteria, small noncoding RNAs (sRNAs) base pair with complementary sites in mRNA targets, to regulate mRNA translation and turnover[1]. sRNA binding rapidly modulates gene expression in response to a variety of physiological and environmental cues and sRNA regulation is essential for stress response and pathogenesis in bacteria[2,3].

Although guide RNAs enable precise recognition of genes, RNA-guided enzymes must search through a large excess of non-specific DNA or RNA to locate the desired target sequence. According to the Smoluchowski model for diffusion in a simplified unhindered environment[4], finding a unique target site through random 3D diffusion should take minutes in bacterial cells[5]. This timescale is incompatible with estimates that bacteria respond to sRNA induction within 1-2 min of receiving an environmental signal[6]. Similar inconsistencies between the observed response and that predicted by 3D diffusion were noted for DNA binding proteins such as *E. coli* lac repressor[7]. In response, theoretical frameworks have been devised to explain enhanced search strategies. The Berg-Winter-von Hippel model proposed an intermittent search process called facilitated diffusion that combines a 3D search and local 1D sampling of DNA or RNA molecules[8]. During facilitated diffusion, optimized partitioning of the

1D and 3D search time speeds up location of a specific binding site, because a protein can scan a large fragment of DNA or RNA before dissociating.

Because the protein components of RNA-guided enzymes typically bind the target as well as the guide RNA, they have the potential to search for targets through some form of facilitated diffusion[9]. For example, 1D diffusion improves the search time by Cas9 and guide RNAs[10]. A scanning mechanism was proposed to allow bacterial RNase E to reach downstream mRNA sites[11]. Therefore, a similar strategy may account for the kinetics of sRNA regulation. *E. coli* sRNAs are chaperoned by Hfq, a unique matchmaker protein capable of facilitating the base-pairing of many different sRNA·mRNA pairs[1,12]. Hfq recognizes its sRNA and mRNA clients through sequence motifs common to the members of each RNA class: the proximal surface of Hfq recognizes 3′ terminal uridines present in the intrinsic terminators of all sRNAs, whereas the distal surface recognizes repeated (ARN)$_n$ trinucleotide motifs present in the major class of mRNA targets[13]. Additionally, six arginine patches on the lateral rim of the Hfq hexamer interact non-specifically with the sRNA body and the mRNA, with a modest preference for U/A or U/C-rich single-strands[14,15]. The arginine patches are essential for sRNA·mRNA annealing by Hfq[16].

[1]Thomas C. Jenkins Department of Biophysics, Johns Hopkins University, 3400 N. Charles St.,5, Baltimore, MD 21218, USA. [2]Present address: Laboratory of Single-Molecule Biophysics, International Institute of Molecular and Cell Biology in Warsaw, Trojdena 4, Warsaw 02-109, Poland. ✉e-mail: emalecka@iimcb.gov.pl; swoodson@jhu.edu

To facilitate sRNA annealing and regulation, Hfq must be recruited to the mRNA, typically through binding to (ARN)$_n$ motifs[17–19]. However, the distances between Hfq binding sites and sRNA binding sites vary substantially amongst mRNA targets of sRNA regulation. In transcriptome-wide crosslinking studies[20,21], Hfq binding sites were usually located close to known sites of sRNA

complementarity, but the distances ranged from 0 to ~60 nt (Fig. 1A). In studies on individual mRNAs, sRNA-mRNA base pairing was still efficient when the Hfq and sRNA binding sites were ≥80 nt apart[22,23]. Moreover, (ARN)$_n$ motifs can be relocated within an mRNA without a significant loss of Hfq-dependent regulation[22]. It is not known how Hfq locates sRNA binding sites at various distances from its own

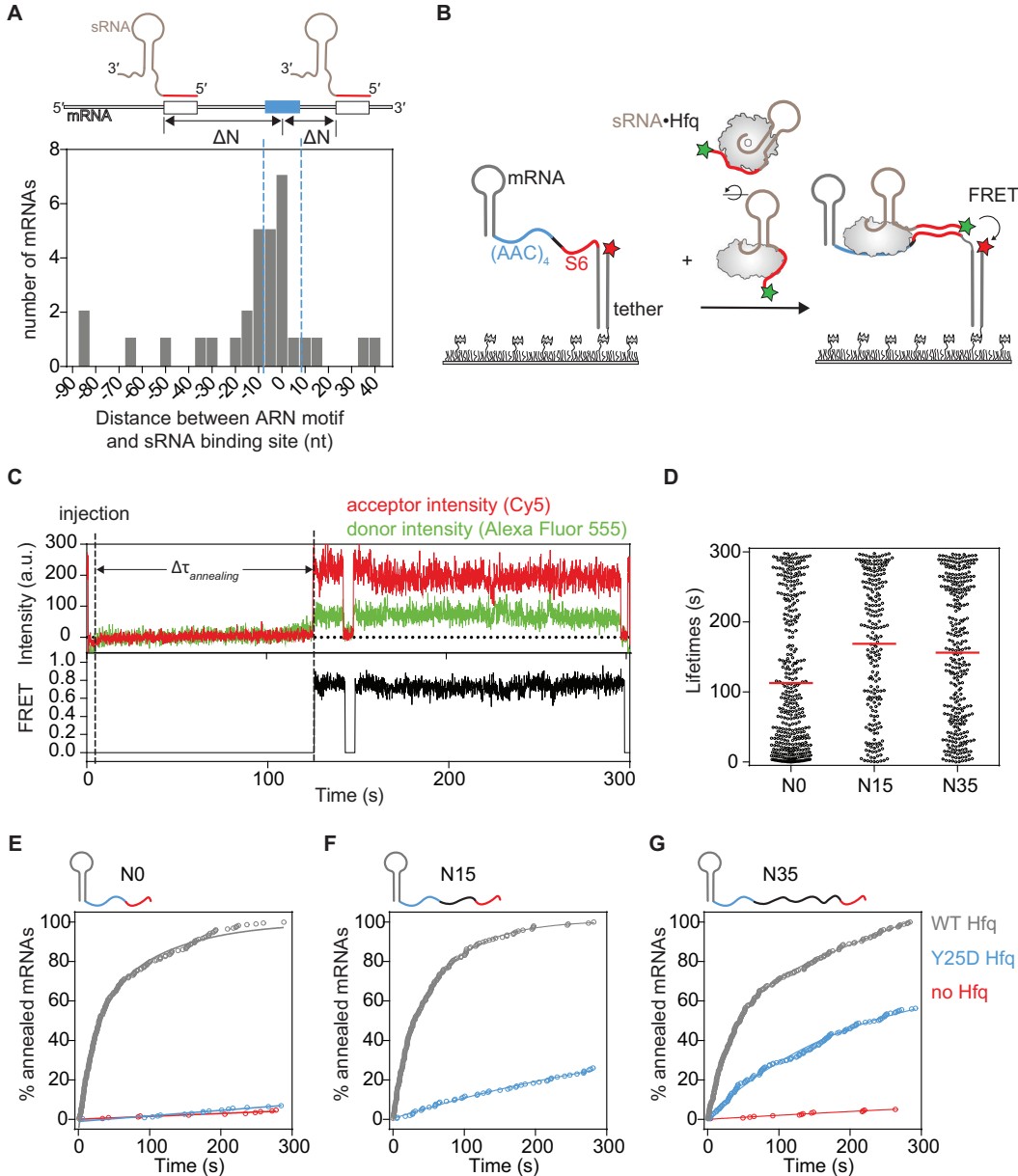

**Fig. 1 | sRNA annealing with mRNA sequences far from the Hfq binding site.**
**A** Variable sequence separation between Hfq ARN recognition site (blue box) and sRNA binding sites (open box) in mRNA targets of sRNA regulation. Histogram of distances between the center of a strong ARN motif (five ARN triplets with up to 2 mismatches) and the 5′ end of an sRNA binding site compiled from CRAC data[21] (Fig. 2J). **B** smFRET assay for sRNA·mRNA annealing. Base pairing between Alexa 555-sRNA and the immobilized mRNA•Cy5-DNA tether complex results in high FRET. Model mRNAs contain an Hfq binding site (AAC)$_4$ (blue), a spacer N (black) and an sRNA binding site S6 (red), followed by 5 nt and the RNA-DNA hybrid. See Supplementary Fig. S1A for details. **C** Representative sRNA binding reaction for a single mRNA with no spacer between (AAC)$_4$ and the sRNA binding site ((AAC)4-N0-S6). Fluorescence intensity (top panel) and FRET (bottom panel) upon 532 nm laser excitation. Cy5 was directly excited in the first 9 and last 10 frames. The moment of sRNA-Hfq injection (dashed line) is marked by slight increase in background

fluorescence. The time from sRNA injection to first annealing event ($\Delta\tau_{annealing}$) is indicated. **D** Distributions of sRNA-mRNA dwell times for mRNAs with different spacer lengths. The mean is marked with a red line. A fraction of binding events persisted until the end of the movie (300 s); ((AAC)4-N0-S6, 32%, $N = 154$; (AAC)4-N15-S6, 49%, $N = 219$; (AAC)4-N35-S6, 63%, $N = 313$). See Supplementary Fig. S1B–D for further data. **E–G** Cumulative distribution of sRNA-Hfq arrival times ($\Delta\tau_{annealing}$, as in **C**), in the presence of wild type Hfq (gray), distal Hfq mutant Y25D (blue), or without Hfq (red) for mRNAs with spacers (E) N0, (F) N15 and (G) N35. Data for WT Hfq were fit with a double exponential equation: N0 (154 events), $k_{fast} = 0.05 \pm 0.001\ s^{-1}$, $k_{slow} = 0.01 \pm 0.002\ s^{-1}$; N15 (219 events), $k_{fast} = 0.06 \pm 0.004\ s^{-1}$, $k_{slow} = 0.015 \pm 0.0006\ s^{-1}$; N35 (313 events), $k_{fast} = 0.03 \pm 0.006\ s^{-1}$, $k_{slow}$ - slow. See Supplementary Fig. S1 for further data. Source data for this and following figures are provided as a Source Data file.

binding site, and whether this is aided by a 1D search across the mRNA itself.

Here, we use single-molecule FRET to investigate how Hfq-sRNA complexes locate target sites within an mRNA. The results show that after Hfq is recruited to the $(ARN)_n$ motif, it brings the sRNA to distant target sites in the mRNA molecule by compacting the mRNAs through interactions with the rim arginines. We also show that Hfq can transfer the sRNA between different target sites without dissociating from the mRNA. When unstable sRNA-mRNA duplexes unzip, the Hfq-sRNA complex rescans the mRNA in search of a more stable sRNA binding site. We propose that this iterative 1D search explains how Hfq and sRNAs flexibly target many different bacterial mRNAs within a short time for effective post-transcriptional gene control. Related forms of 1D scanning may operate in other RNA complexes.

## Results

### Hfq locates sRNA binding site away from its interaction site on an mRNA

Using single-molecule FRET, we previously showed that sRNA-mRNA annealing on Hfq is reversible and occurs in distinct steps[24]. To study how the distance between Hfq- and sRNA-binding sites within a target mRNA (Fig. 1A) affects the base-pairing efficiency, we designed a model sRNA and various mRNAs that mimic the features of natural Hfq substrates, similar to those used before[24]. The model sRNA contains a 5′ seed region that pairs with the target site in the mRNA, and 3′ U-rich motifs that bind the rim and proximal face of Hfq (Supplementary Fig. S1A). The mRNAs carry upstream AAC repeats, which are recognized by the distal surface of Hfq, and a downstream target site complementary to the sRNA seed. The complementary region was 6 bp long, which is the minimal duplex length that supports regulation of *ptsG* mRNA by SgrS sRNA[25]. The model mRNAs used in this study were named after the number of AAC repeats, the length of the spacer between the Hfq and sRNA binding sites (N) and the number of complementary bases (S) in the sRNA binding site (e.g., (AAC)4-N0-S6).

To monitor the kinetics of sRNA-mRNA base pairing using total internal reflection fluorescence (TIRF) microscopy, the mRNA was immobilized on a passivated quartz slide via a biotinylated, Cy5-labeled complementary DNA tether (Fig. 1B). sRNA labeled with Alexa Fluor 555 was injected into the channel as a 5 nM 1:1 complex with Hfq hexamer. After injecting the sRNA-Hfq complex, we observed frequent colocalization of Alexa 555 and Cy5 fluorophores, denoting recruitment of the sRNA to the immobilized mRNA (Fig. 1C). In the absence of mRNA, sRNA binding to the slide was negligible[24]. The FRET efficiency of the colocalized fluorophores was high ($E \sim 0.76$, Supplementary Fig. S1B, middle panel), consistent with a fully base-paired duplex that brings the attached fluorophores near each other. Most high FRET complexes were stable, with a mean lifetime ≥112 s (Fig. 1D). The true lifetime is longer, because 32% of events persisted until the end of the movie (5 min, Supplementary Fig. S1B, top panel).

The distribution of sRNA arrival times on (AAC)4-N0-S6 mRNA (Fig. 1C, $\Delta\tau_{annealing}$) was best described by a fast observed rate of $0.05\,s^{-1}$ (49%) and a slower process ($0.01\,s^{-1}$) (Fig. 1E, gray line), in agreement with our previous observations[24]. Little sRNA annealing was observed in the absence of Hfq (Fig. 1E, red line, and ref. 24).

We next asked whether separation of the Hfq and sRNA binding sites affects the sRNA annealing kinetics by comparing mRNAs with 0-, 15- and 35-nt spacers between the $(AAC)_4$ motif and the sRNA-binding site ((AAC)4-N15-S6 and (AAC)4-N35-S6, respectively). The spacer sequences were random and were not enriched for Hfq recognition motifs (Supplementary Table S1–3) or predicted to form secondary structure[26]. The 35-nt spacer was comparable to the separation between Hfq and sRNA binding sites in natural mRNA targets (Fig. 1) and was the longest sequence we could design that was not predicted to form stable secondary structure. All three mRNAs formed stable complexes with the sRNA, with average lifetimes >150 s (Fig. 1D).

The high FRET efficiency of these complexes (Supplementary Fig. S1C, D, middle panels) indicated that the sRNA seed was able to pair with the specific complementary site in the mRNA, even when placed at a distance from the Hfq binding site. This result indicated that there must be some mechanism for overcoming the separation between Hfq and sRNA binding sites that does not require dissociation of the sRNA-Hfq complex.

Recruitment of Hfq to the $(AAC)_4$ motif in the 5′ half of the mRNA was needed for efficient sRNA-mRNA annealing, even if the Hfq- and sRNA-binding sites are far apart. A Y25D mutation on the distal face of Hfq that impairs binding to the $(AAC)_4$ site reduced annealing of the $N = 0$ mRNA to the level without Hfq (Fig. 1E, blue line). Some binding to the longer mRNAs was still observed (Fig. 1F, G and Supplementary Fig. S1), however, suggesting that the spacer helps recruit Hfq. This could occur through short-lived interactions of adenines with the distal surface of Hfq, or solely on rim interactions, making extensive contact with the distal surface unnecessary. For all mRNAs, however, the annealing rate was significantly slower with the Y25D mutant than with wild type Hfq (Fig. 1G), confirming the importance of the $(AAC)_4$ motif.

### Hfq compacts RNA through the interactions with the rim

Since Hfq must be recruited to the $(AAC)_n$ motif for efficient annealing, we wondered how the sRNA can base pair with complementary sequences distant from this motif. It is possible that the $(AAC)_n$ motif initially recruits the Hfq-sRNA complex but is later released as Hfq scans the mRNA spacer via nonspecific interactions (Fig. 2A). Alternatively, the distal face of Hfq may remain bound to the $(AAC)_n$ motif while its other RNA binding surfaces scan the mRNA. The mRNA conformation should not appreciably change in the first scenario, whereas the mRNA should appear more compact in the second scenario (Fig. 2A).

To directly observe changes in the mRNA conformation, we designed an mRNA with the Cy5 acceptor inserted downstream of an $(AAC)_6$ motif and a single-stranded region (35 nt spacer plus 6 nt sRNA site plus 6 nt linker) between Cy5 and Cy3 attached to the 3′ DNA tether (hereafter (AAC)6-Cy5-N35-S6). Changes in the end-to-end distance of the RNA between the two fluorophores were monitored through changes in FRET efficiency (Fig. 2A). In our standard buffer (50 mM Tris-HCl pH 7.5, 50 mM NaCl, 50 mM KCl), the FRET efficiency was low ($E \sim 0.22$, Fig. 2B), confirming that the spacer does not form a stable secondary structure.

A FRET histogram constructed from data obtained over the first minute after injecting 0.5 nM WT Hfq showed two peaks centered at $E \sim 0.18$ and $E \sim 0.58$ (Fig. 2C, Supplementary Fig. S2C). Two-dimensional histograms of the binding trajectories showed that the mRNA remains in a low FRET state until Hfq binds, corresponding to the peak at $E \sim 0.18$ (Supplementary Fig. S2A). After Hfq binds, the mRNA adopts a mid-FRET conformation, corresponding to the peak at $E \sim 0.58$ (Supplementary Fig. S2A). Although the mid-FRET conformation was often stable (Supplementary Fig. S2B), individual mRNAs occasionally visited lower FRET structures (Fig. 2C). Because these low FRET transitions were more prevalent at 5 nM Hfq (Fig. 2D; $E \sim 0.3$), they may represent transfer of the mRNA to a second Hfq hexamer[27]. Although we do not know the structure of the mRNA spacer when bound to Hfq, these results showed that Hfq binding brings the 3′ sRNA target site closer to the upstream Hfq binding site. This compaction of the mRNA is consistent with previous SAXS experiments and an smFRET study of Hfq bound to OxyS sRNA[14,28–30].

We next tested which binding surfaces of Hfq are required to compact the mRNA, by using Hfq variants with mutations in the distal or rim RNA binding sites. In our single molecule assay, the distal mutant Hfq:Y25D also compacted the mRNA spacer (Fig. 2E), demonstrating that specific binding to the AAC motif is not essential for this change in mRNA conformation although it is required for efficient

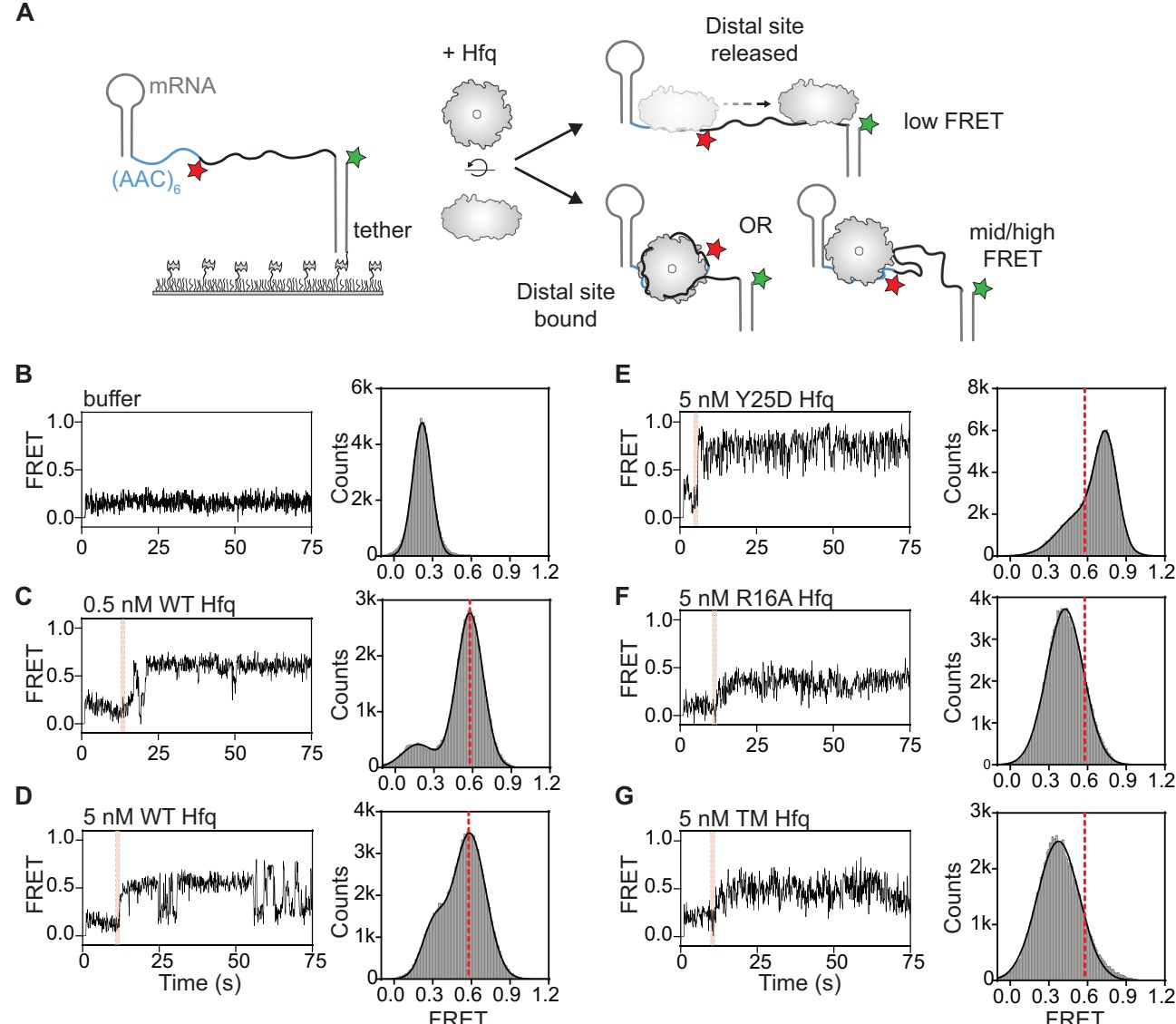

**Fig. 2 | Hfq compacts RNA through interactions with the rim. A** smFRET experiment to monitor conformational changes in the mRNA upon Hfq binding. If Hfq leaves the AAC site and scans downstream, then FRET will be low. If Hfq remains on the AAC site while scanning, then FRET will increase. Cy5 was inserted downstream from (AAC)$_6$; Cy3 was attached to the tether DNA. In (AAC)6-Cy5-N35-S6 mRNA the fluorophores are separated by 46 nt single-stranded nucleotides. Six AAC triplets were used to ensure strong Hfq binding in the absence of sRNA. **B–G** (Left) Representative FRET trajectories and (Right) FRET values for immobilized dual-labeled mRNA (AAC)6-Cy5-N35-S6 upon injection of indicated Hfq variants; the pink stripe indicates the moment of Hfq binding. See Supplementary Fig. S2C for the fluorescence intensities. The red dashed line indicates the mean FRET efficiency of the WT Hfq complex. **B** RNA before injection of Hfq ($N = 159$ molecules); **C–G** FRET values over the first minute after Hfq injection (see Methods). **C** 0.5 nM wild type Hfq ($N = 145$ molecules), **D** 5 nM wild type Hfq ($N = 134$ molecules), **E** 5 nM distal Y25D Hfq mutant ($N = 335$ molecules), **F** 5 nM rim R16A Hfq mutant ($N = 234$ molecules), **G** 5 nM triple-rim R16A, R17A, R19D Hfq mutant ($N = 145$ molecules).

sRNA-mRNA annealing. Interestingly, Hfq:Y25D binding produced higher average FRET efficiency ($E ≈ 0.75$) than WT Hfq ($E ≈ 0.58$). Perhaps Hfq:Y25D can scan the mRNA more freely because it is not anchored to the (AAC)$_6$ motif. By contrast, Hfq variants with a substitution in one rim arginine (Hfq:R16A) or all rim arginines (Hfq:R16A, R17A, R19D or 'TM') compacted the mRNA spacer less than WT Hfq did, yielding an average $E ≈ 0.43$ (R16A, Fig. 2F) and $E ≈ 0.37$ (TM, Fig. 2G). These results showed that arginines on the rim of Hfq are needed to bring the 5' and 3' ends of the mRNA closer together.

### Hfq compacts RNA homopolymers

To find out if Hfq-mediated compaction is sequence-specific, we used a similar smFRET strategy to test the effect of Hfq on 50 nt rU and rC homopolymers[31,32] (Fig. 3A). In the absence of Hfq, the FRET efficiency

was low ($E ≈ 0.13–0.17$) consistent with coiling of the single-strands (Fig. 3B, C, top panels). When Hfq was added to immobilized rU$_{50}$, a subset of RNAs transitioned to a higher FRET structure, $E ≈ 0.75$, with subsequent visits to intermediate FRET structures, $E ≈ 0.3–0.6$ (Fig. 3B, Supplementary Fig. S3B). This change, which is very similar to that observed for our designed mRNA, can be explained by the known interaction of single-stranded U-rich sequences with the rim of Hfq[15]. No changes in FRET efficiency were observed upon the addition of Hfq to immobilized rC$_{50}$ (Fig. 3C, Supplementary Fig. S3C), which does not interact with Hfq in EMSA experiments (Supplementary Fig. S3A).

To assess whether basic residues of Hfq simply neutralize the RNA charge, allowing it to coil or base pair more tightly, we washed the immobilized RNAs with buffer containing 300 mM NaCl in the absence of Hfq. The FRET efficiency of (AAC)6-Cy5-N35-S6 mRNA increased

substantially in 300 mM NaCl ($E \sim 0.65$; Supplementary Fig. S3D) compared to 50 mM NaCl ($E \sim 0.18$; Fig. 2B), suggesting that the mRNA can form some secondary structure in high salt. By contrast, the FRET efficiency of the homopolymers changed very little in 300 mM NaCl (Supplementary Fig. S3E, F), consistent with previous estimates of the effect of salt on the persistence length of single-stranded RNA[33]. Thus, Hfq compacts single-stranded RNA more potently than can be explained by charge neutralization alone. From these results, we inferred that the RNA ends are brought together because of how they interact with Hfq and not by general salt effects.

## mRNA flexibility provides the communication between Hfq- and sRNA-binding sites

The results above show that Hfq can shorten the distance between the 5′ and 3′ ends of the mRNA by folding or looping the spacer into a more compact structure. A flexible spacer would be important if Hfq remains anchored to the ARN motif while searching for sequences complementary to the sRNA. The next question was whether this folding is needed to bring the sRNA seed sequence closer to distant target sites. To test this possibility, we blocked folding of the internally labeled mRNA ((AAC)6-Cy5-N35-S6, Fig. 4A) using a 28 nt antisense oligonucleotide (ASO) that hybridizes to the spacer between (AAC)$_6$ and the sRNA binding site (Fig. 4B). After hybridization with the mRNA, excess free ASO was removed to reduce the likelihood of non-specific interactions. We confirmed that ASO hybridization extended the distance between the dyes, reducing the FRET efficiency ($E \sim 0.09$, Supplementary Fig. S4A) relative to the mRNA alone ($E \sim 0.22$, Fig. 2B). The addition of Hfq alone to the mRNA-ASO hybrid did not affect the distance between fluorophores (Supplementary Fig. S4B), showing that the protein cannot fold the spacer into a more compact structure when it is double-stranded.

To test the effect of the blockade on sRNA annealing efficiency, we immobilized the mRNA on the slide via an unlabeled DNA tether and monitored the binding of 5′-Alexa Fluor 555-sRNA in complex with Hfq. In the absence of the ASO, most of the complexes were stable for several minutes ($\tau_{long} = 370 \pm 50$ s, Fig. 4C, E, Supplementary Fig. S4C). A small fraction of events ($a = 0.15 \pm 0.05$) were short-lived ($\tau_{short} = 1.3 \pm 0.3$ s). This distribution between short-lived and stable binding changed dramatically when folding of the mRNA was prevented by the ASO (Fig. 4D). Most of the observed complexes existed transiently ($\tau_{transient} = 0.3 \pm 0.03$ s, $a_{transient} = 0.5 \pm 0.03$) or were short-lived ($\tau_{short} = 4 \pm 0.3$ s $a_{short} = 0.29 \pm 0.02$), indicating little base pairing with the sRNA site. A minority of encounters produced long-lived complexes ($\tau_{long} = 198 \pm 44$ s, $a_{long} = 0.22 \pm 0.04$, Fig. 4D, E, Supplementary Fig. S4D). These observations show that Hfq searches for distant sRNA binding sites through folding or looping of the flexible segments between the (AAC)$_n$ and sRNA sites.

## Hfq bridges distal binding motif and sRNA-mRNA duplex

The results above show that Hfq is recruited to the mRNA through its distal surface (Fig. 1E–G) and transfers the sRNA to a distant binding site, which requires looping of the intervening RNA. To address whether Hfq detaches from the AAC motif when the sRNA and mRNA base pair, we first inspected the FRET efficiency of complexes between the internally labeled mRNA (AAC)6-Cy5-N35-S6 and 5′ end-labeled sRNA (from Fig. 4A), which reflects the distance between the Hfq binding site and the sRNA-mRNA duplex. Control experiments confirmed that sRNA-mRNA duplexes are fully base paired and (AAC)$_6$ motif does not change the sRNA binding lifetimes significantly (Supplementary Fig. S4E). Because the fluorophores are 40 nt apart ($N = 35$), the sRNA-mRNA complex should occupy a low FRET state. Instead, we observed a wide distribution of FRET efficiencies, with average $E \sim 0.39$ (Fig. 4F, Supplementary Fig. S4F), suggesting that Hfq holds the sRNA-mRNA complex in a more compact state. This compact state depended on Hfq, because Cy3-labeled DNA hybridized to the sRNA site without Hfq

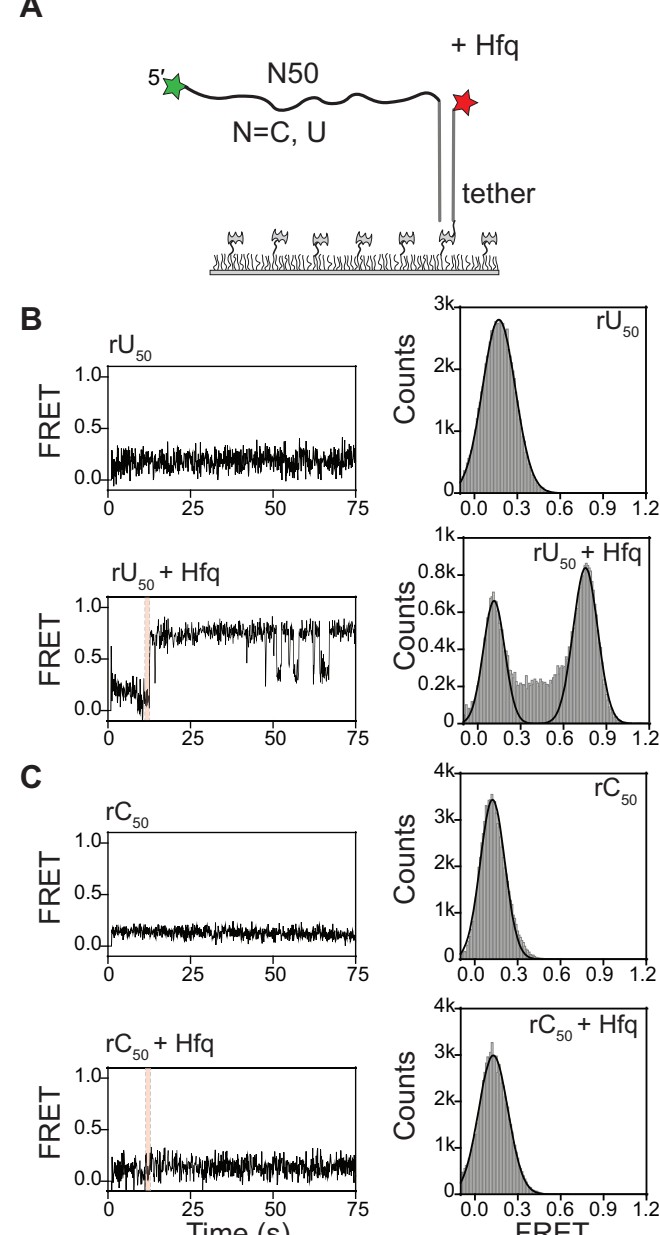

**Fig. 3 | Hfq compacts polyuridine RNA. A** FRET efficiency monitors the end-to-end distance of rU$_{50}$ and rC$_{50}$ single-stranded RNA in the presence of Hfq. **B**, **C** (Left) Representative FRET trajectories for immobilized RNAs, with and without Hfq. The pink stripe marks the injection of 5 nM wild type Hfq. (Right) The distribution of FRET values (**B**) for rU$_{50}$ before injection (top, $N = 148$ molecules), over the first minute of injection (bottom, $N = 80$ molecules), **C** for rC$_{50}$ before injection (top, $N = 132$ molecules), over the first minute of injection (bottom, $N = 142$ molecules). Two peaks in the histogram for rU$_{50}$ result from separate populations rather than transitions between compact and extended states in individual molecules (see also Supplementary Fig. S3B, C).

resulted in a much lower FRET efficiency ($E \sim 0.17$; Fig. 4G, Supplementary Fig. S4G).

Next, we directly checked whether Hfq continues to simultaneously contact the (AAC)$_6$ motif and the sRNA-mRNA duplex after annealing, acting as a molecular bridge. If this is the case, the (AAC)$_6$ motif should remain occupied by Hfq and inaccessible for base-pairing with a TTG ASO complementary to the (AAC)$_6$ motif (see Supplementary Table S1). To measure AAC accessibility, the immobilized

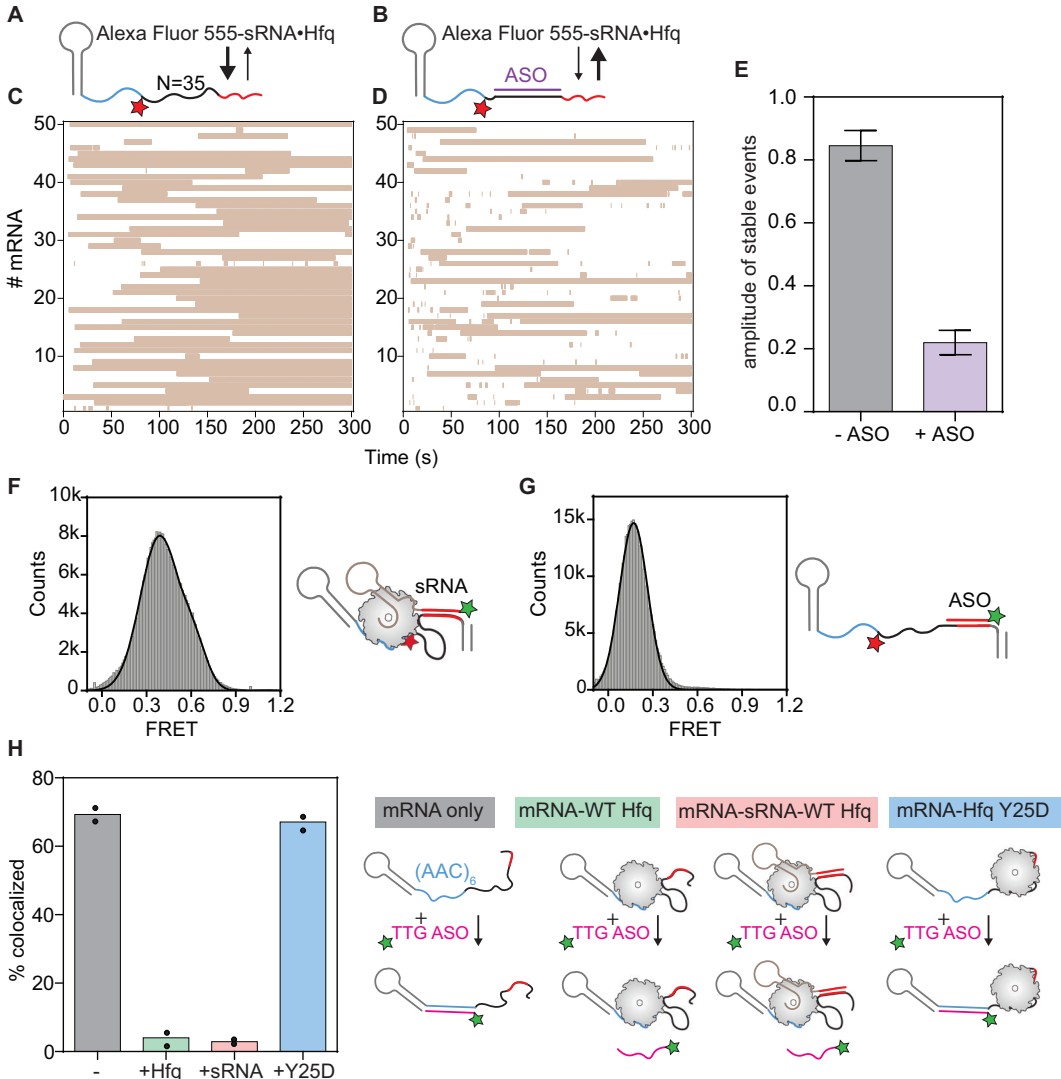

**Fig. 4 | mRNA flexibility is required for Hfq to anneal the sRNA with a distant target site.** **A**, **B** smFRET monitors the binding of Alexa Fluor 555-labeled sRNA to immobilized mRNA (AAC)6-Cy5-N35-S6. **A** The 35 nt ssRNA spacer can be coiled by Hfq (see Fig. 2). **B** When the spacer is stiffened by hybridizing with a 28 nt antisense oligomer (ASO), it is not compacted by Hfq (see Supplementary Fig. S4A, B). Control experiments with Cy3-labeled ASO showed that 84.5 ± 0.3% mRNAs bind the ASO ($N$ = 420 molecules). **C**, **D** Rastergrams depicting sRNA-Hfq interactions with 50 randomly selected single (AAC)6-Cy5-N35-S6 mRNAs. Each horizontal bar represents a single binding event. **C** −ASO; **D** +ASO. **E** Proportion of stable sRNA complexes, from maximum likelihood analysis of the dwell times of sRNA-mRNA binding. Error bars represent the s.d. of the mean (bootstrap); − ASO, $N$ = 322 events; +ASO, $N$ = 752 events. See Supplementary Fig. S4C, D for details of the fits and statistics. **F** FRET efficiency of sRNA-mRNA complexes in the presence of Hfq ($N$ = 322 binding events). See also Supplementary Fig. S4F. **G** FRET of an 11 nt 5′

Cy3-ASO binding to the same mRNA without Hfq. The 5′ end of the oligomer pairs with the 3′ end of S6 (red), whereas the ASO 3′ end extends beyond S6 by 5 nt to ensure duplex stability in the absence of Hfq. See also Supplementary Fig. S4G. **H** The mRNA (AAC)₆ Hfq binding site is inaccessible in Hfq and sRNA complexes. Accessibility is based on the fraction of immobilized Cy5-labeled mRNAs in the field of view that hybridize with a Cy3-TTG ASO probe. mRNAs were incubated with buffer (gray), 0.5 nM wt Hfq (green), 5 nM wt Hfq-sRNA complex (red), 5 nM Y25D Hfq (blue). Unbound Hfq and sRNA were washed out after 5 min and the mRNAs were incubated with 10 nM Cy3-TTG-ASO for an additional 5 min. The percentage of Cy5-labeled mRNAs bound by Cy3-TTG-ASO was then calculated for each condition: gray (69 ± 3%, $N$ = 597 molecules), green (4 ± 1%, $N$ = 720 molecules), red (2.9 ± 0.2%, $N$ = 728 molecules), blue (67 ± 2%, $N$ = 701 molecules). Symbols represent the values of two independent measurements; bar indicates the mean.

Cy5-labeled mRNA was incubated with Cy3-labeled TTG ASO for 5 min and the co-localization fraction was scored. For the free mRNA, ~70 ± 3% colocalized with Cy3-DNA after 5 min (Fig. 4H, gray bar). When the mRNA was pre-incubated with 0.5 nM Hfq, only 4 ± 1% mRNAs were accessible to the TTG ASO (Fig. 4H, green bar). When the mRNA was pre-incubated with the sRNA-Hfq complex, only 2.9 ± 0.2% of immobilized mRNAs could still bind the TTG ASO (Fig. 4H, red bar). Thus, the AAC motif continues to be occupied by Hfq in the annealed sRNA-mRNA complex. The Hfq:Y25D distal face mutant with impaired binding to the AAC motif did not reduce ASO colocalization,

confirming that ASO binding reports specific recognition of the AAC motif (Fig. 4H, blue bar).

## Hfq transfers sRNA between binding sites on mRNA

The inspection of natural mRNAs that are subject to sRNA regulation[34] showed that many contain cryptic binding sites for their cognate sRNA, in addition to the target site that base pairs stably with the sRNA (Supplementary Fig. S5A). The need to search among cryptic sites raised the possibility that sRNAs transfer between sites while Hfq remains bound to the mRNA (Fig. 4H). If an unstable sRNA-mRNA

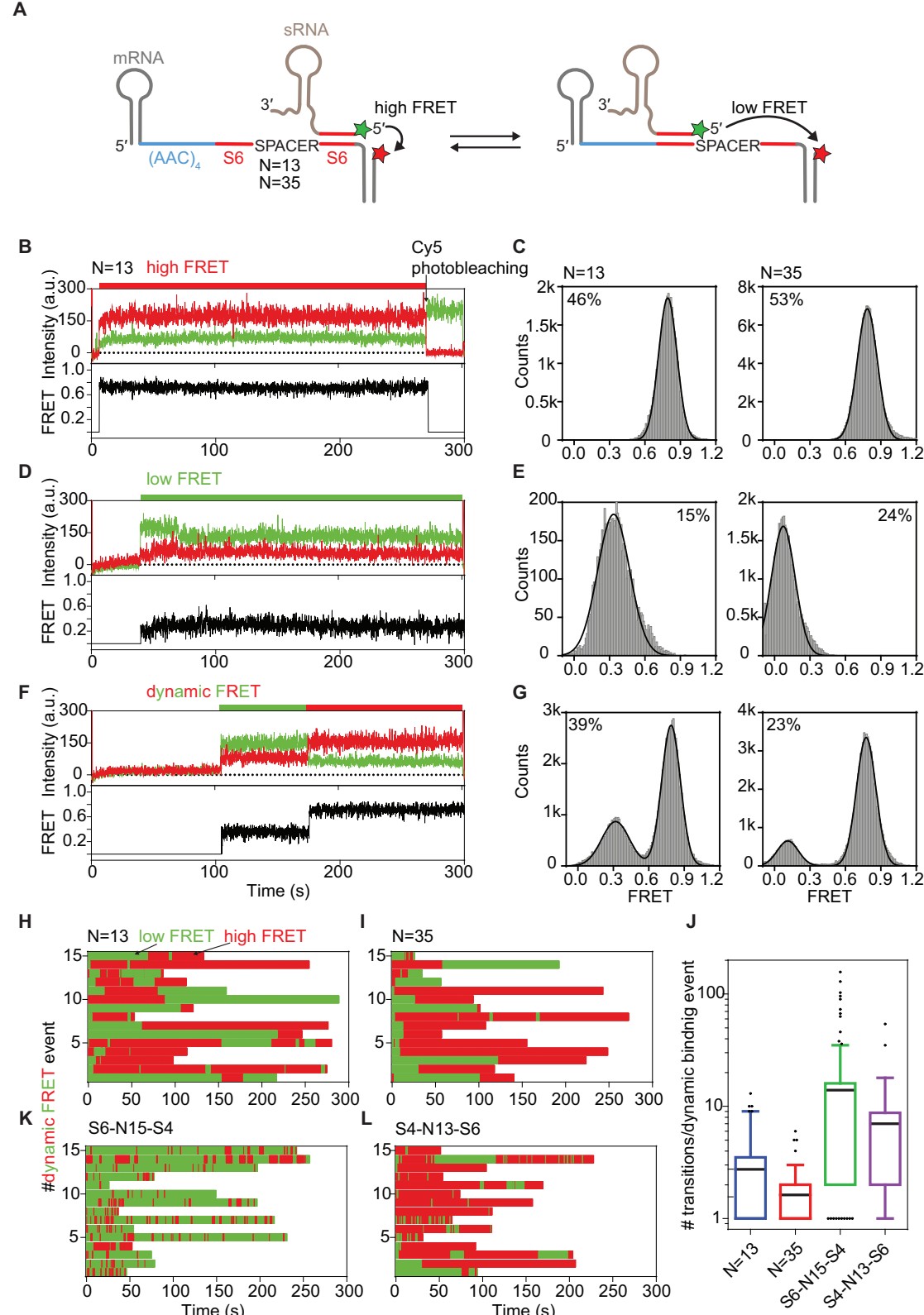

duplex unzips, the ternary complex does not dissociate. Instead, the sRNA attempts to base pair with another site in the same mRNA. This repetitive annealing process would speed the search for the optimal sRNA binding site in the vicinity of an ARN motif.

We reasoned that such intra-strand transfer is likely faster than the time resolution of our method (0.1 s) since we did not observe a delay in reaching a high FRET state when the mRNA contained a single sRNA

binding site (Fig. 1C). Therefore, to investigate whether Hfq can transfer an sRNA between sites in the same mRNA without dissociating from it, we designed mRNAs with tandem target sites to trap potential shuttling events[35,36] (Fig. 5A). The tandem mRNAs contain a single $(AAC)_4$ motif and two identical 6 nt-long sRNA binding sites separated by a short ($N = 13$) or long ($N = 35$) single-stranded spacer ($(AAC)4$-S6-N13-S6 and $(AAC)4$-S6-N35-S6, respectively, Supplementary

**Fig. 5 | Hfq transfers sRNAs between binding sites in a single mRNA. A** Tandem mRNA containing two identical sRNA binding sites (6 nt each) traps sRNA shuttling. Binding to the site adjacent to the $(AAC)_4$ motif results in a low FRET state, while binding to the far site yields a high FRET state. **B, D, F** Representative trajectories of Cy3-sRNA binding to a tandem mRNA with a 13 nt spacer between the sRNA binding sites $((AAC)4-S6-N13-S6)$, illustrating different outcomes. **B** sRNA binds to the distant site (high FRET). **D** sRNA binds adjacent to the $(AAC)_4$ motif (low FRET). **F** sRNA transfers from a low to a high FRET site. Dynamic binding events start with low FRET more often than high FRET: $86 \pm 3\%$ of dynamic sRNA binding events for $(AAC)_4$-S6-N13-S6 and $86 \pm 4\%$ for $(AAC)_4$-S6-N35-S6. **C, E, G** FRET histograms of sRNA binding events to a tandem mRNA with (left) 13 nt spacer ($N = 127$ events) or (right) 35 nt spacer ($N = 218$ events). Percentages are the fractions of binding events assigned by ebFRET to a (**C**) constant high FRET state (**E**) constant low FRET state, (**G**) dynamic FRET with more than 1 transition between low and high FRET states. See Fig. S5I-L for the number of molecules used. **H, I, K, L** FRET rastergrams for dynamic sRNA binding events that result in transfer between sites. **H** $N = 13$, **I** $N = 35$, **K** S6-N15-S4, **L** S4-N13-S6. The events are post-synchronized to the first frame when the fluorescence intensity increased due to sRNA-Hfq co-localization with the immobilized mRNA. Green – low FRET, red – high FRET state. See also Supplementary Fig. S4 B, D, F, G. **J** Number of sRNA transitions between two 6 nt target sites for $N = 13$ (blue) and $N = 35$ (red), and between a 4 nt and 6 nt site (S6-N15-S4, green and S4-13-S6, purple). The whiskers are drawn from the 10th to the 90th percentile; box indicates 1st to 3rd quartile. Mean number of transitions per mRNA (black horizontal line): 2.8 for $N = 13$ ($N = 49$ events), 1.6 for N = 35 ($N = 50$ events), 16.2 for S6-N15-S4 ($N = 114$ events), 7.4 for S4-N13-S6 ($N = 28$ events).

Tables S2 and 3). The sRNA site further from the $(AAC)_4$ motif is closer to the acceptor fluorophore on the DNA tether, and Alexa Fluor 555-sRNA binding to this distant site yields the same high FRET state as the single-site mRNA ($E \sim 0.79$, Fig. 5B, C). The second sRNA binding site is closer to the $(AAC)_4$ motif and further from the acceptor fluorophore. sRNA binding to this site results in a lower FRET state ($E \sim 0.34$ for $N = 13$ or $E \sim 0.07$ for N = 35, Fig. 5D, E).

When the sRNA-Hfq complex was added to the tandem mRNA, we observed both stable low FRET and high FRET complexes (Fig. 5B, D). Surprisingly, high FRET events denoting sRNA binding further from the $(AAC)_4$ motif were more populated (46–53% of total events) than low FRET events (15–24%) indicating that Hfq easily scans the mRNA and prefers annealing the sRNA to a distant site. This preference may result from the mRNA wrapping with arginine patches, strain created by base pairing adjacent to the AAN motif, or residues flanking the target site, which are typically 3′ adenosine in natural mRNAs[34]. As the sites in our tandem mRNAs are followed by U or G, respectively, we expect that the impact of the flanking sequence is minimal. Furthermore, we observed shuttling between low FRET and high FRET states for 39% of observed sRNA encounters with $N = 13$ mRNA (Fig. 5F, G, Supplementary Fig. S5B). Because these transitions occurred without a loss of FRET signal, a single sRNA can transfer between target sites without dissociating from the mRNA.

The FRET changes occurred in both directions (high to low and low to high), indicating that the Hfq-sRNA complex can go back and forth on the mRNA in search of an optimal sRNA binding site (Fig. 5H). When the spacer was 13 nt long ($N = 13$), we observed 2.8 transitions per dynamic binding event on average (Fig. 5J). The average lifetimes of the sRNA annealing events with the tandem target mRNA ($24.6 \pm 4.7$ s and $58.6 \pm 8.7$ s, respectively, Supplementary Fig. S5C) were shorter than the lifetimes of annealed complexes containing only one sRNA binding site (>100 s, Fig. 1D). Because the sRNA-mRNA complementary regions have the same sequence, they are both equally likely to be unzipped by Hfq. Therefore, the apparent stability of the single sRNA-mRNA duplex suggests that unzipping events are too transient to be detected unless the sRNA can be trapped on a second target site. A similar observation has been made for Ago-guide complexes, which are also able to shuttle between two target sites on ssDNA[36].

Shuttling was also observed for the mRNA with a longer spacer, $N = 35$ (Supplementary Fig. S5D), but the fraction (23%) and the average number of transitions per binding event (1.6) decreased in comparison to the $N = 13$ spacer (Fig. 5I, J). The lower number of transitions does not stem from a shorter overall lifetime of the ternary complex leading to sRNA transfer, since the total Hfq•sRNA•mRNA binding time is comparable to mRNA $N = 13$ (Supplementary Fig. S5E). Thus, the longer spacer seems to raise the kinetic barrier for sRNA shuttling between target sites, perhaps because the longer spacer has some internal structure or makes additional interactions with Hfq.

We next asked whether shuttling allows the sRNA to seek its optimal binding site within an mRNA. In experiments with tandem

mRNA variants containing a 4 nt (S4) and a 6 nt (S6) target site, the sRNA predominantly occupied the longer S6 site regardless of its position relative to the $(AAC)_4$ motif (Fig. 5K, L, Supplementary Fig. S5F, G). Less stable base pairing with S4 resulted in frequent hops from the S4 site to the S6 site, increasing the number of transitions per dynamic mRNA binding event (Fig. 5J). This observation indicated that the frequency of sRNA transfer within an mRNA depends on the stability of sRNA-mRNA base pairing, directing the sRNA to complementary targets. Typically, sRNA binding sites found in nature are longer than the 6 base pairs that were examined in this study. Consequently, we investigated whether the strength of sRNA-mRNA interactions rises as the length of the complementary region increases. The results revealed that the average lifespans of paired complexes progressively lengthened when the duplexes were 8 and 10 base pairs long (Supplementary Fig. S5H). This suggests that once the sRNA establishes a stable pairing, the likelihood of shuttling also diminishes.

## sRNA transfer on mRNA depends on the Hfq rim

Since the R16A substitution on the rim of Hfq causes the protein to scan the RNA less actively (Fig. 2F, G), we wondered whether this mutation also impairs sRNA annealing to sites distant from Hfq. When WT Hfq-sRNA complexes were added to the tandem mRNA with $N = 35$, the sRNA spends ~80% of the time in the distant high FRET state (Fig. 6A, B, red bar, Supplementary Fig. S5I). In the presence of R16A Hfq, however, many attempts to transition from the nearby (low FRET) site to the more favorable distant site were unsuccessful, resulting in the sRNA reaching only a mid-FRET state followed by a return to the low FRET state (Fig. 6C). In total, the sRNA spent half as much time on the distant site in the presence of the Hfq rim mutant compared to the WT protein (Fig. 6B, green vs red bars, S5K). This result showed that arginines on the lateral rim of Hfq not only compact the mRNA but allow the Hfq-sRNA complex to search for stable sRNA binding sites over longer distances.

Because the rim of Hfq mediated the compaction of $rU_{50}$ but not $rC_{50}$ (Fig. 3B, C), we next tested whether the Hfq-sRNA complex can scan across a C-rich linker to reach the distant target site. We constructed a tandem mRNA containing 30 cytosines between the two sRNA sites $((AAC)4-S6-N30C-S6)$. Replacing the original spacer with cytosines reduced the time spent on the distant sRNA binding site even further, to 25% (Fig. 6B, purple bar, Supplementary Fig. S5L). Therefore, a search across C-rich sequences is inefficient even when Hfq is stably recruited to the $(AAC)_4$ motif. Overall, these results suggest that Hfq most efficiently scans, and compacts sequences enriched in uridines, presumably because these sequences interact more favorably with RNA binding sites on the rim of the protein.

Finally, we asked whether Hfq can bypass secondary structure when shuttling between sRNA binding sites. We constructed a tandem mRNA with a spacer containing 13 single-stranded nt plus a stem-loop $((AAC)4-S6-N13-S6\_SL$ or $N = 13\_SL$ for short, Supplementary Fig. S6A). The sRNA was able to transition between the two binding sites yielding low and high FRET states, as anticipated (Supplementary Fig. S6B).

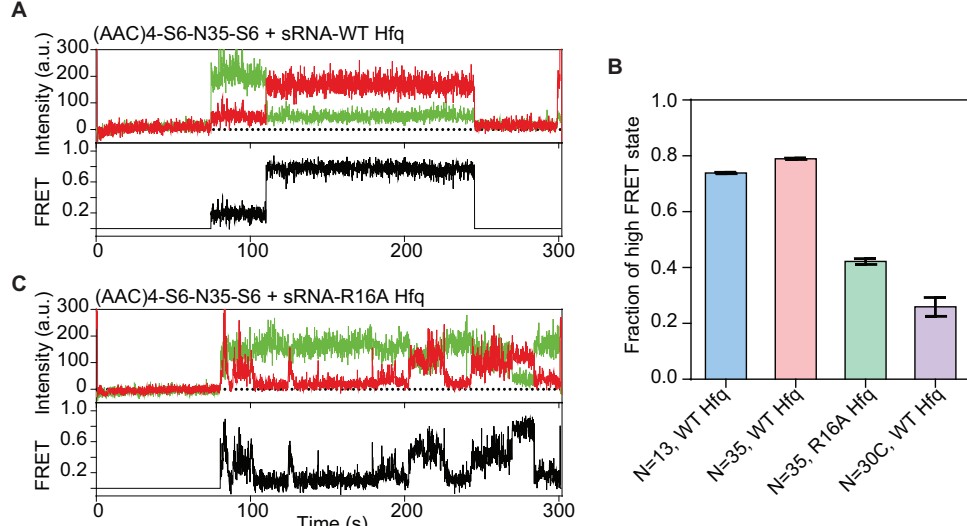

**Fig. 6 | sRNA transfer on mRNA depends on the Hfq rim.** Representative trajectory of sRNA binding to a tandem mRNA with a 35 nt spacer between sRNA binding sites in the presence of (**A**) wt Hfq (**C**) Hfq:R16A. **B** The fraction of time the sRNA resides in a high FRET on the indicated mRNAs, calculated from population densities of FRET values (Supplementary Fig. S5H–K). Bars represent mean ± SD of a binomial distribution.

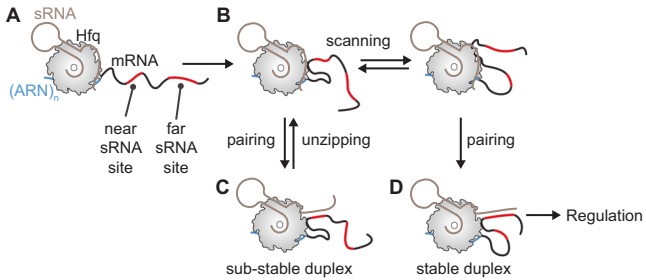

**Fig. 7 | RNA scanning enables an iterative search for stable sRNA-mRNA pairs.** Proposed pathway for target recognition in an mRNA containing an A-rich binding site for Hfq. **A** The distal surface of Hfq engages an (ARN)$_n$ motif in the mRNA (blue). **B** Adjacent nucleotides interact with arginine patches on the rim of Hfq, which preferentially bind single-stranded uridines. One-dimensional diffusion or hopping allows different segments of the mRNA to contact the Hfq rim, compacting the mRNA while Hfq's distal face remains anchored to the (ARN)$_n$ motif. **C** The arginine patches facilitate nucleation of sRNA-mRNA base pairs at regions of complementarity (red). Unstable duplexes unzip, allowing a new sRNA-mRNA duplex to form. Interrogation of the mRNA sequence continues without dissociation of the Hfq-sRNA-mRNA complex. **D** When the sRNA-Hfq complex encounters an optimal binding site, a stable sRNA-mRNA duplex is formed, leading to the regulation of mRNA expression.

Notably, the fraction of dynamic binding events increased from 39% for $N = 13$ to 57% for $N = 13\_SL$ (Supplementary Fig. S6C). Even though the sRNA binds these mRNAs with similar average lifetimes (Supplementary Fig. S6D), the average number of transitions per dynamic binding event increased from 2.5 to 4 (Fig. SE), corresponding to a slightly shorter dwell time of the sRNA in the high FRET state (Supplementary Fig. S6F). Thus, a stem-loop does not impede mRNA scanning.

## Discussion
### Iterative search for optimal sRNA targets
During the search for regulatory targets, sRNAs must sort through many transcripts containing candidate sites that are partially complementary to the sRNA seed region. Our results show how Hfq accelerates the target search to match cognate sRNA-mRNA pairs

within the 1–2-minute window for sRNA regulation in bacteria (Fig. 7). Firstly, diffusion-limited binding of Hfq to (ARN)$_n$ motifs in the mRNA ($k_{on} \sim 1$–$10 \cdot 10^7 \, M^{-1} s^{-1}$ [27,37];) recruits sRNAs to plausible candidates for regulation, narrowing the search to sRNA and mRNA pairs bearing Hfq binding motifs. Secondly, we show that Hfq compacts flexible RNA segments, bringing sites far from the ARN motif to the rim of Hfq, where they can base pair with the sRNA. The need to compact the RNA may explain why the sRNA initially pairs with sites closest to Hfq (Fig. 5). Thirdly, we propose that repeated zipping and unzipping of sRNA-mRNA duplexes leads to the kinetic selection of optimal target sites within the vicinity of an Hfq binding site. In support of this idea, we show that Hfq can transfer an sRNA between sites on a single mRNA without dissociating from it. As the transfer rate depends on the stability of the sRNA-mRNA duplex, the sRNA preferentially resides on target sites that are fully complementary to the sRNA seed sequence. Finally, our smFRET results show that Hfq bridges the two RNAs in the sRNA-Hfq-mRNA complex. This allows for further rounds of scanning, base pairing, and duplex unzipping, until the sRNA-Hfq-mRNA complex finally dissociates (Fig. 7).

### Arginine-mediated scanning of RNA
RNA compaction and sRNA annealing to distant target sites require arginine residues on the lateral rim of Hfq (Fig. 2 and Fig. 6) that are also required for Hfq's annealing activity[16]. Indeed, small angle X-ray scattering studies revealed that *rpoS* mRNA adopted a more extended conformation with the rim mutant Hfq:R16A than with WT Hfq[14], like the model mRNA tested here (Fig. 2F, G). Alanine substitutions of the rim arginines also abrogated helix initiation by Hfq[38]. The structures of Hfq-RNA complexes suggest how these basic residues may mediate a 1D search for sRNA target sites. In *E. coli* Hfq, R16, R17 and R19, K47 and R66 form a positively charged strip along the outer edge of each subunit in the Hfq hexamer that preferentially binds single-stranded RNA[12]. Because the interactions with each arginine are chemically similar, the target RNA may slide past the rim, presenting different residues to the sRNA for base pairing. In this model, nucleotides between the rim and the ARN motif form a loop that shrinks or grows, depending on which residues are engaged on the rim. Interestingly such a loop is observed in 3'ETS$^{leuZ}$ RNA bound to Hfq PNPase[39]. Another possibility is that RNA segments interact with successive arginine patches along the outer rim of Hfq[15]. This "spooling" would

also compact the RNA but requires that the RNA cross over the flexible C-terminal domains, which has not been observed yet.

The rim of Hfq has been shown to preferentially recognize U-rich single-strands[15,40]. Because $rU_{50}$ was strongly compacted in our experiments (Fig. 3), we hypothesize that Hfq scans bound RNAs by successively interacting with uridine clusters that are present in most transcripts. Interestingly, Hfq-bound mRNA fragments from transcription-wide studies were enriched in uridines as well as ARN motifs[20,21,41]. For example, in 5′ UTR peaks bound by Hfq[20], we noticed that $U_4$ and $U_5$ tracts are 2.3- and 3.6-fold enriched, respectively, compared to random sequences. Since the spacer in our model mRNA did not contain long stretches of uridines, yet was still scanned by Hfq, shorter uridine clusters may suffice for this purpose. Moreover, not all Hfq-bound peaks identified by crosslinking contained ARN sequences[20]. Since we observed that the distal Hfq:Y25D mutant also compacted the mRNA (Fig. 2E), it is possible that rim interactions can also help locate ARN motifs.

The arginine-mediated scanning model predicts that the Hfq R16A mutation will have little effect on the regulation of targets possessing overlapping or adjacent sRNA and Hfq binding sites, whereas this mutation should be deleterious for sRNA-target pairs with separated sRNA and Hfq-binding sites. This trend was in fact observed by[42], who measured the impact of Hfq:R16A on a set of well-studied sRNA-mRNA pairs in *E. coli* (Supplementary Table S4). It is important to acknowledge that R16 plays a pivotal role in the annealing process as a whole[16], thus complicating the interpretation of Hfq mutations in vivo. Moreover, this comparison only covers a small set of sRNA-mRNA pairs. Nevertheless, the scanning model broadly agrees with the results of these in vivo reporter assays.

### RNA restructuring

Hfq binding was previously shown to compact *rpoS* and *fhlA* mRNAs and OxyS sRNA[14,28,29]. In these examples, simultaneous interactions between the ARN motif and distal face of Hfq and between U-rich sequences and the rim of Hfq were required to fully restructure the RNA. It is possible that compaction or looping of our model mRNA (Fig. 2) led to the formation of secondary structures that are otherwise unstable, in agreement with early studies showing that structural changes induced by Hfq in *ompA* mRNA were maintained after proteolytic digestion of the protein[43]. Hfq-induced folding of certain RNAs may explain why Hfq-crosslinking studies[20,21] and proximity ligation experiments[41,44] recovered RNA sequences longer than the minimal Hfq binding sites. Hfq-induced folding can also lead to the direct regulation of mRNA expression. It has been shown that Hfq sequesters A- and U-rich patches in *cirA* mRNA and represses its translation[45]. *Pae*Hfq inhibits *amiE* translation by interacting with an extended A-rich sequence, with the help of the auxiliary protein Crc that stabilizes Hfq-*amiE* mRNA interactions[46].

### Comparison to search strategies used by other RNA-guided systems

Other RNA-guided enzymes use various mechanisms of facilitated diffusion to speed their target search, such as sliding (constant association), hopping (micro-dissociation) and intersegmental transfer (jumping between motifs close in space)[47,48]. Single-molecule FRET experiments showed that *Clostridium butyricum* Ago (CbAgo) can slide short distances on ssDNA targets or transfer to another ssDNA segment to bypass large obstacles such as proteins[36]. Human argonaute 2 (hAgo2) was also shown to shuttle across short distances (~20 nt) on a target RNA, although the mechanism of diffusion was not determined[35]. Unlike Hfq, Ago proteins only recognize their targets through base pairing with the guide RNA. In this sense, Hfq is more similar to Cas proteins that recognize the protospacer adjacent motif (PAM) in the DNA target before DNA melting and guide RNA-DNA pairing. Interestingly, nonspecific interactions between positively

charged residues in Cas9 and the DNA backbone away from the PAM (post-PAM) were recently proposed to facilitate DNA binding before PAM recognition[49,50]. This use of non-specific interactions with the target is reminiscent of the Hfq rim-mediated search. However, PAM sequences are adjacent to the target, while Hfq's action requires more flexibility to reach sRNA target sites far from its own recognition sequence. This unique capability is made possible by stable anchoring of the distal Hfq surface on ARN motifs while neighboring RNA segments are interrogated by the sRNA in association with an arginine patch on the rim.

Other RNA chaperones and processing enzymes must also have the capacity to slide or hop along an RNA. We propose that arginine patches are a universal mechanism for achieving exchangeable interactions that allow RNAs to move across protein surfaces. For example, the ring-shaped Ro protein from *Deinococcus radiodurans* is lined with arginine and lysine side chains that bind small Y RNAs and contribute to its chaperone properties[51]. In another example, the exosome directs RNA substrates through its central channel toward its nuclease sites[52]. Structural studies showed that the RNA-binding path is lined with arginines that bind RNA weakly, allowing the RNA to move along the channel[53]. Finally, arginine-rich proteins are known to contribute to exchangeable RNA interactions within RNP condensates and granules[54]. In Hfq, RNA movement is coupled to the zipping and unzipping of base pairs between the sRNA and mRNA target, facilitating the search for optimal targets of bacterial sRNA regulation.

## Methods
### RNA and DNA preparation

Synthetic DNA and RNA oligonucleotides were purchased from IDT or Thermo Fisher and are listed in Supplementary Tables S1 and S2. DNA templates for in vitro transcription were prepared by extension of overlapping oligonucleotides (Supplementary Table S2) using Q5 polymerase (NEB). RNAs were transcribed using phage T7 RNA polymerase in the presence of 1 mM NTPs, 40 mM DTT, 30 mM $MgCl_2$, and 1× transcription buffer (80 mM Tris-HCl pH 8, 2 mM spermidine, 10 mM NaCl). RNAs were purified on 8 M urea 8% polyacrylamide gels, eluted overnight in 0.3 M Na-acetate pH 5.4, 1 mM EDTA, and precipitated with ethanol. The pellets were dissolved in water. The sequences of in vitro transcripts are listed in Supplementary Table S3.

For 5′ labeling with fluorophores, sRNA was transcribed with the addition of 32 mM GMP. The resulting 5′-monophosphate was treated with EDC (1-ethyl-3-[3-dimethylaminopropyl]carbodiimide hydrochloride) and imidazole[55], then coupled with Alexa Fluor 555 NHS succinimidyl ester in carbonate-bicarbonate buffer pH 8.5 according to the manufacturer's instructions. The modified RNA was purified with Chroma TE-10 spin columns (Takara Bio), ethanol precipitated and dissolved in water. RNA labeling efficiency was close to ~100%. DNA oligomers were purchased with a 5′-amino linker C6 (Thermo Fisher Scientific) and modified with Cy3 NHS ester (Cytvia) in carbonate-bicarbonate buffer pH 8.5 according to the manufacturer's instructions.

Internally labeled mRNA (AAC)6-Cy5-N35-S6 (Supplementary Table S3) was constructed by splint ligation of a chemically synthesized 5′ RNA fragment (5′mRNAyWT+C) and a 3′ RNA fragment generated by T7 in vitro transcription in the presence of 32 mM GMP. The DNA template for transcription of the 3′ fragment was generated from overlapping oligomers Cy5-6BP + 35INS-3part_F and Cy5-6BP + 35INS-3part_R. The 5′ RNA fragment (0.5 nmol) and monophosphorylated 3′ RNA fragment (1 nmol) were annealed with a DNA splint (Cy5-6BP-35INS_splintLig2, 0.6 nmol) in 75 μL 20 mM KCl by heating to 95 °C for 40 s, followed by cooling to 25 °C (−1 °C/40 s). Ligation was then carried out in a total volume of 375 μL (50 mM Tris-HCl pH 7.5, 0.4 mM ATP, 2 mM $MgCl_2$, 1 mM DTT, 50 mM KCl, 180 U T4 RNA ligase 2) for 1 h at 25 °C. The reaction was stopped by the addition of EDTA to 2 mM final concentration, followed by purification on an 8 M urea 8% polyacrylamide gel.

## Protein expression and purification

Untagged *E. coli* K12 Hfq, Hfq:Y25D, Hfq:R16A, and Hfq:TM proteins were over-expressed in BL21(DE3)Δ*hfq::cat-sacB* cells and purified by metal chelation and size exclusion chromatography as previously described[24]. 'Hfq' throughout the manuscript refers to its hexamer.

## Electrophoretic mobility gel shift assay

Cy3-labeled $U_{50}$ and $C_{50}$ RNAs for binding assays were annealed with Cy5-labeled RNA tether[31] in 1×TNK buffer supplemented with 5% glycerol. RNAs were mixed 1:1 and annealed either by heating to 75 °C followed by refolding at 37 °C and incubation at 20 °C or heating to 80 °C for 2 min followed by slow cooling to 20 °C (−1 °C/min) (see Supplementary Fig. S3A). Hfq binding reactions were carried out in a 10 μL volume, with 20 nM RNA and 40 or 80 nM Hfq (final concentration), for 15 min at RT. Samples were loaded onto a pre-run native 8% polyacrylamide gel in 1×TBE and run at 15 W for 1.5 h at 4 °C. The gels were imaged using a GE Typhoon scanner.

## Single-molecule data acquisition

Single-molecule data were recorded using a custom-built prism-based total internal reflection fluorescence (TIRF) microscope[56] with an EMCCD camera (Andor). A 532 nm laser and a 633 nm laser were used for Cy3/Alexa Fluor 555 and Cy5 excitation, respectively. Each movie (100 ms/frame) started with 9 frames of 633 nm excitation to localize Cy5-labeled mRNAs in the field of view. Next, movies were recorded for 5 min with 532 nm excitation to monitor FRET or colocalization between Cy3/Alexa Fluor 555 and Cy5 fluorophores. At the end of the movie, 1 s 633 nm excitation reported on Cy5 photobleaching.

## Single-molecule experiments

Designed mRNAs (Supplementary Table S3) were immobilized on the slide via a biotinylated DNA tether (Bio-SA5) conjugated to Cy3, Cy5 dyes or without a fluorophore, as indicated in the text. Homopolymers were immobilized through an 18 nt Cy5-labeled biotinylated RNA[31]. Immediately before immobilization, 20 nM DNA tether, 60 nM RNA were annealed by denaturation at 75 °C for 5 min in 1×TNK buffer, refolded at 37 °C for 15 min and equilibrated at 20 °C. The reaction also contained 160 nM antisense oligomer where indicated in the text. Samples were diluted 50-fold in imaging buffer (10 mM Tris-HCl, pH 7.5, 50 mM NaCl, 50 mM KCl, 4 mM Trolox, 0.01% octaethylene glycol monododecyl ether (Nikkol), 0.8% glucose, 2 U RNasin Plus) and immobilized on quartz slides coated with DDS and pretreated with biotinylated BSA (0.2 mg/ml), Tween-20 (0.2%) and Neutravidin (0.1 mg/ml). Unbound heteroduplexes were washed with imaging buffer supplemented with an oxygen scavenging system (OSS, 165 U/ml glucose oxidase, and 2170 U/ml catalase) to reduce the photobleaching[57].

Prior to experiments, sRNA was heated to 75 °C for 5 min followed by refolding at RT for 10–15 min. sRNA-Hfq complexes were incubated in 1× TNK buffer for 5–15 min at RT at 100 nM final concentration. Immediately before use, complexes were diluted to 1 or 5 nM final concentration (as indicated in the text) in imaging buffer with OSS and injected into the flow channel. To note, observed events reflect specific sRNA-mRNA interactions, as no significant binding events were observed without mRNA substrate (Małecka and Woodson, 2021). For experiments with Hfq alone, the final protein concentration is indicated in the figures. The Hfq solution was flowed into the channel at the ~150th frame.

## Single-molecule data quantification and statistical analysis

The donor and acceptor channels were mapped, and the time trajectories of single molecules were extracted from CCD images using custom IDL code[58]. Cy5 molecules were automatically selected based on spots appearing in the acceptor channel during the first 10 frames of the movie. Trajectories were then processed using MATLAB.

Background subtraction and leakage correction were performed as described[58]. All experiments presented in the study are combinations of two to four individual trials.

**FRET histograms.** The FRET efficiency was calculated from Eq. (1), in which $I_D$ and $I_A$ are the baseline and leakage-corrected emission intensities of the donor and acceptor, respectively.

$$E_{FRET} = \frac{I_A}{I_A + I_D} \tag{1}$$

FRET histograms were produced from the FRET efficiency in individual frames. In experiments testing the effect of Hfq on RNA conformation, the histograms were composed of data from the 150th frame (Hfq injection) to the 750th frame. To check the Alexa Fluor 555-labeled sRNA base-pairing status with mRNA, FRET histograms were prepared from frames containing active fluorophores (excluding frames after Cy5 photobleaching). FRET histograms were constructed and fitted with Gaussian distributions in GraphPad Prism. For sRNA shuttling experiments with tandem mRNAs, the fraction of time $p$ that an sRNA spent in the low or high FRET state was calculated from the area $A$ under the curve for each peak in the distribution, $f_i = p_i/(p_{low} + p_{high}) = A_i/(A_{low} + A_{high})$.

**Hidden Markov Model analysis.** Specific frames in the fluorescence trajectories were assigned to FRET states using ebFRET[59] available on GitHub repository: https://github.com/ebfret. After leakage and background corrections and removal of photobleaching, the frames with sRNA binding events were extracted and used as an input for the analysis. After assigning states to the binding events, the dwell times for sRNAs residing in the low or high FRET state and the number of transitions per binding event were extracted.

**Dwell-time analysis.** Dwell times of Hfq•sRNA residence on the mRNA were collected using custom MATLAB code. To estimate the lifetimes and associated errors, unbinned distributions were fit using a maximum likelihood estimation[60] to equations containing two or three exponential terms (Eqs. (3) and (4)). In all equations, $t_m$ is the minimum resolvable time interval in the experiment, $t_x$ represents the duration of the experiment, $\tau_1, \tau_2, \tau_3$, represent characteristic lifetimes and $a, a_1,$ and $a_2$ are the fitted amplitudes.

$$\frac{1}{a \times \left(e^{-\frac{t_m}{\tau_1}} - e^{-\frac{t_x}{\tau_1}}\right) + (1-a) \times \left(e^{-\frac{t_m}{\tau_2}} - e^{-\frac{t_x}{\tau_2}}\right)} \times \left(\frac{a}{\tau_1} \times e^{-\frac{t}{\tau_1}} + \frac{1-a}{\tau_2} \times e^{-\frac{t}{\tau_2}}\right) \tag{3}$$

$$\frac{1}{a_1 \times \left(e^{-\frac{t_m}{\tau_1}} - e^{-\frac{t_x}{\tau_1}}\right) + a_2 \times \left(e^{-\frac{t_m}{\tau_2}} - e^{-\frac{t_x}{\tau_2}}\right) + (1-a_1-a_2) \times \left(e^{-\frac{t_m}{\tau_3}} - e^{-\frac{t_x}{\tau_3}}\right)}$$
$$\times \left(\frac{a_1}{\tau_1} \times e^{-\frac{t}{\tau_1}} + \frac{a_2}{\tau_2} \times e^{-\frac{t}{\tau_2}} + \frac{1-a_1-a_2}{\tau_3} \times e^{-\frac{t}{\tau_3}}\right) \tag{4}$$

Errors in the fitted parameters were determined by bootstrapping 1000 random samples of initial data, fitting the resultant values with a normal distribution, and determining the standard deviation, σ. The error for $a_3$ was obtained by propagation of errors in $a_1$ and $a_2$, assuming $a_1 + a_2 + a_3 = 1$. Histograms visualizing the maximum likelihood fits were generated by unequal binning of the data. Error bars in the histograms represent the standard deviation $\sigma = \sqrt{(NP(1-P))}$, where N is the number of events and P is the event probability.

## Reporting summary

Further information on research design is available in the Nature Portfolio Reporting Summary linked to this article.

## Data availability

The data reported in this paper and any additional information required to reanalyze them are available from the Johns Hopkins Research Data Repository at https://doi.org/10.7281/T1/YPUEQT. Source data are provided with this paper.

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

## Acknowledgements
The authors thank Drs. L. Ganser and S. Myong for the gift of fluorophore-labeled homopolymer RNAs. This work was supported by a grant from the National Institute of General Medicine [R35 GM136351-03 to S.A.W.]

## Author contributions
E.M.M. designed and performed the experiments, analyzed data and wrote the manuscript; S.A.W. reviewed and editing the manuscript and acquired funding.

## Competing interests
The authors declare no competing interests.
