## [Peer Review File · Nature Communications]

RNA compaction and iterative scanning for small RNA targets
by the Hfq chaperoneREVIEWER COMMENTS

Reviewer #1 (Remarks to the Author):

Small RNAs (sRNA) are an important class of posttranscriptional regulator in bacteria that often work in concert with the RNA chaperone Hfq. They frequently act to repress the translation of target mRNA species by base-pairing with them close to the Shine-Dalgarno sequence. Pairing between the sRNA and the mRNA involves a seed sequence on the sRNA which can be as short as 6 nt. Pairing between the seed on the sRNA and the complementary sequence on the mRNA target is promoted by Hfq which can simultaneously bind the sRNA and target mRNA using different surfaces of the molecule. Binding of Hfq to the mRNA target can occur through a specific binding site on the RNA consisting of ARN motifs that Hfq binds through its so-called distal surface. Target mRNAs therefore often contain both a specific binding site for Hfq and a target site that pairs with the sRNA seed region, with the distance between these sites varying anywhere between being close or relatively far away (up to 80 or more nt). This interesting and thorough study by Malecka and Woodson uses elegant single-molecule FRET assays to address how Hfq-bound sRNAs locate their target sites on mRNAs. Importantly, they present convincing new evidence that Hfq facilitates the location of target sites that are distant from the ARN motif by a 1D search across the mRNA. This search (i) only occurs on sequences enriched in uridines, (ii) is facilitated by arginine patches present on the rim of the hexameric Hfq molecule, and (iii) results in compaction of the RNA. They also show that for a target mRNA that contains a single ARN motif but two target sites for the sRNA, Hfq can transfer the sRNA between target sites without dissociating from the mRNA. This transfer occurs in a manner that is again dependent on the rim arginines. These findings are significant as they help explain how Hfq facilitates the target search of those sRNAs it binds. All of my comments and questions are minor.

Specific Comments:

1. In Figure 1 the authors use a Hfq mutant containing substitution Y25D to make the case that Hfq binding to the ARN motif is important for efficient pairing of the sRNA with targets positioned 0, 15, and 35 nt downstream of the ARN motif (as the Hfq mutant cannot efficiently recognize this motif). However, the pairing efficiency of the sRNA with the target is higher with the Hfq(Y25D) mutant using the 35 nt spacer when compared to the 15 nt spacer. Not all target mRNAs contain an ARN motif, so does this result suggest that ARN-independent access to a target mRNA might play a more significant role for targets that are positioned further away from an ARN site?
2. I was hoping the authors could comment in the discussion on whether structure present between the ARN motif and the sRNA target site would potentially interfere with the scanning of an sRNA by Hfq. Or could Hfq potentially hop a structure? As the authors point out, some natural mRNAs have been reported to have target sites that are 80 nt away from the ARN motif and would thus potentially have some degree of structure between the ARN motif and target site.
3. The experiments in Figure 5 indicating that Hfq can enable an sRNA to switch between two target sites positioned downstream of an ARN motif, employed a construct with two identical target sites. It was somewhat surprising that the target site closest the ARN motif seemed to be targeted less frequently by the sRNA than the one positioned further away because this site might be expected to be encountered first in a 1D search. The authors suggest the preferential filling of the more distal site is explained by formation of a more stable complex when this site is filled, perhaps by wrapping of the RNA with the arginine patches. However, can the authors rule out the possibility that the observed occupancy preferences aren't due to sequences flanking the target sites that might facilitate greater pairing with the sRNA in one case when compared to the other?

Reviewer #2 (Remarks to the Author):

Małecka et. al elucidate the molecular mechanism of Hfq-sRNA complex dynamics with an associated mRNA molecule. The authors use single-molecule FRET to show that Hfq-sRNA can locate sRNA binding site several nucleotides away from Hfq binding site on an mRNA strand. They demonstrate that Hfq compacts the RNA strand to access the sRNA binding site and the rim interactions are necessary for the compaction. They also show sequence dependency of this compaction in that the presence of uridines between the two binding sites are essential to effectively compact the RNA.

Further investigation by the authors shows that blocking the region between the (AAC)_n and the sRNA binding site prevents Hfq-sRNA in accessing the sRNA binding site, demonstrating the importance of the flexible segments between the two sites. Using a similar strategy, the authors also show the importance of accessibility of (AAC) site and Hfq acts as a bridge between the two binding sites. In the final set of experiments, the authors use an mRNA with tandem target sites separated by spacers to observe if Hfq shuttles between the two sites. They notice that the complex shuttles between the two target sites with the lifetimes of each target site binding shorter than that of a single binding site mRNA. It is also noticed that the shuttling probability is reduced once stable base pairing is ensued. Lastly, the authors show that the rim of the protein not only help in compaction but also help in the search of sRNA binding sites.

Based on these experiments, the authors propose a stepwise model in which Hfq-sRNA initially binds the ARN motif on the mRNA. Subsequently, the arginine residues in the rim of the protein facilitate in scanning for the complementary region to the sRNA. The scanning happens without dissociation of the protein complex and once the binding site is found, it leads to a stable duplex of sRNA-mRNA.

Overall, the experiments are well-designed, extensive, and well-thought-out. The authors have systematically endeavored to elucidate the mechanism of target finding and recognition of Hfq. This manuscript stands out as one of the most exemplary Hfq papers published recently, presenting a compelling model. It is a solid paper deserving publication in Nature Communications with only minor corrections required. I have only a few minor questions that I would appreciate the authors addressing.

Major comments

None

Minor comments

1. Fig S1B (WT Hfq) shows a small percentage of low-FRET peak which is absent/not pronounced in the other histograms. Is it donor-only peak or some other artefact?
2. In line 129-130, it is mentioned that a spacer helps recruit Hfq. Could the authors elaborate on this? Is this a consequence of using a particular spacer sequence or is it a general statement?
3. In Fig. S3A, the bands are denoted with "•Hfq" - does it mean that Hfq was added, or is it complexed?
4. In line 195, the authors explain how mRNA flexibility is essential for Hfq target search. Please elaborate how the flexibility is different from the accessibility of the mRNA. Can their method decouple the two different contributions?
5. In the same section, (AAC)₆ motif is used instead of (AAC)₄. Wouldn't that increase the binding lifetime of Hfq and therefore make the ASO increasingly inaccessible to the (AAC) site? In other words, having (AAC)₄ would make the comparison with other experiments fairer, unless the authors show/explain that it does not make much of a difference.
6. Line 280 - The average lifetime(s) of sRNA annealing events for the tandem targets are quite drastic. Is there a particular reason why one of the targets (the one farther from (AAC) site) shows more than twice the lifetime of the other one? Does it have to do with the instability of potential short loops close to the (AAC) site? If so, the frequency of sRNA transfer (Line 299) depends not only the base-pairing but also the 'flexibility' of the mRNA.

Reviewer #3 (Remarks to the Author):

In this study, Malecka and Woodson used single molecule FRET with purified components in vitro to examine how the RNA chaperone protein assists in scanning an mRNA for a site of small RNA base pairing. The authors document that Hfq promotes compaction of the mRNA and also allows the sRNA to test binding to two different sites while remaining bound to the mRNA. These Hfq activities require the arginine residues on the rim of the Hfq hexamer.

The study is very carefully performed, and the manuscript is clearly written. The findings help to clarify how the Hfq protein can facilitate the rapid sRNA-mediated regulation observed in vivo. I only have editorial comments:

1. The following sentences were not optimally clear.

--Page 2, lines 52-53: Clarify that the scanning mechanism is being suggested for RNase E.

--Page 4, line 101: The authors should be clear that when they are always referring to the Hfq hexamer (somewhere in the text).

--Page 5, lines 129-130: The sentence "More events were observed for the longer mRNAs" seems disconnected from the sentences before and after.

--Page 6: line 155-157: I thought more explanation could be provided for the sentence beginning "These low FRET transitions..."

--Page 8, line 206: "cannot fold the double-stranded spacer" was slightly cryptic.

--Page 10, lines 272-273: Why is annealing to the distant site preferred?

--Page 10, line 274: Possibly replace "sites" with "states".

--Page 10, lines 282-284: Sentence beginning "This observation suggests that there are unresolved ..." was slightly cryptic.

2. The authors need to double check their citations (some are numbers, some list the authors).

Response to reviewer critiques

We thank the reviewers for their positive assessment of the work, and for their helpful suggestions for improving the manuscript. Our responses are given below in red type. --EM and SW.

Reviewer #1:

Small RNAs (sRNA) are an important class of posttranscriptional regulator in bacteria that often work in concert with the RNA chaperone Hfq. They frequently act to repress the translation of target mRNA species by base-pairing with them close to the Shine-Dalgarno sequence. Pairing between the sRNA and the mRNA involves a seed sequence on the sRNA which can be as short as 6 nt. Pairing between the seed on the sRNA and the complementary sequence on the mRNA target is promoted by Hfq which can simultaneously bind the sRNA and target mRNA using different surfaces of the molecule. Binding of Hfq to the mRNA target can occur through a specific binding site on the RNA consisting of ARN motifs that Hfq binds through its so-called distal surface. Target mRNAs therefore often contain both a specific binding site for Hfq and a target site that pairs with the sRNA seed region, with the distance between these sites varying anywhere between being close or relatively far away (up to 80 or more nt). This interesting and thorough study by Malecka and Woodson uses elegant single-molecule FRET assays to address how Hfq-bound sRNAs locate their target sites on mRNAs. Importantly, they present convincing new evidence that Hfq facilitates the location of target sites that are distant from the ARN motif by a 1D search across the mRNA. This search (i) only occurs on sequences enriched in uridines, (ii) is facilitated by arginine patches present on the rim of the hexameric Hfq molecule, and (iii) results in compaction of the RNA. They also show that for a target mRNA that contains a single ARN motif but two target sites for the sRNA, Hfq can transfer the sRNA between target sites without dissociating from the mRNA. This transfer occurs in a manner that is again dependent on the rim arginines. These findings are significant as they help explain how Hfq facilitates the target search of those sRNAs it binds. All of my comments and questions are minor.

Specific Comments:

1. In Figure 1 the authors use a Hfq mutant containing substitution Y25D to make the case that Hfq binding to the ARN motif is important for efficient pairing of the sRNA with targets positioned 0, 15, and 35 nt downstream of the ARN motif (as the Hfq mutant cannot efficiently recognize this motif). However, the pairing efficiency of the sRNA with the target is higher with the Hfq(Y25D) mutant using the 35 nt spacer when compared to the 15 nt spacer. Not all target mRNAs contain an ARN motif, so does this result suggest that ARN-independent access to a target mRNA might play a more significant role for targets that are positioned further away from an ARN site?

Yes, we cannot rule out such an explanation. The spacer contains a few single adenosines and an AAA that conceivably establish short-lived contacts with the distal surface of Hfq, potentially enhancing the search through the rim. Another scenario is that a fraction of successful searches may rely solely on rim interactions, rendering extensive contact with the distal surface unnecessary, as suggested by the reviewer. We added this explanation on Page 5. This hypothesis aligns with *in vivo* observations in which some of the Hfq and sRNA targets detected by proximity ligation were not near ARN motifs (Page 14).

2. I was hoping the authors could comment in the discussion on whether structure present between the ARN motif and the sRNA target site would potentially interfere with the scanning of an sRNA by Hfq. Or could Hfq potentially hop a structure? As the authors point out, some natural mRNAs have

been reported to have target sites that are 80 nt away from the ARN motif and would thus potentially have some degree of structure between the ARN motif and target site.

Yes, previous ensemble experiments¹ suggested that Hfq can “hop” secondary structure. To address the reviewer’s question more directly, we added the results of experiments using a new tandem mRNA variant with a spacer containing 13 single-stranded nucleotides plus a stem-loop ((AAC)₄-S6-N13-S6_SL or N=13_SL for brevity, Fig. S6A). The sRNA was able to transfer between two binding sites in N=13_SL mRNA, yielding low and high FRET states ($E_{\text{low}} = 0.28$ and $E_{\text{high}} = 0.79$), as anticipated (Fig. S6B). Notably, we observed sRNA transfer for 57% of binding events to N=13_SL, slightly higher than 39% for N = 13 mRNA (Fig. S6C). Even though dynamic ternary complexes on N=13_SL mRNA have similar lifetimes as ternary complexes formed with N=13 or N=35 mRNAs (Fig. S6D), the average number of transitions per sRNA binding event increased from 2.5 to 4 (Fig. S6E). This faster rate of transfer correlated with a shorter dwell time of the sRNA in the high FRET state, possibly due to the one-nucleotide distance between the stem-loop and sRNA target site (Fig. S6F). These findings suggest that a stem-loop does not impede mRNA scanning. A stem-loop could be bypassed if the mRNA detaches from the rim of Hfq and rebinds in another register, while the ARN motif remains bound to the distal face of Hfq. Alternatively, the base of the stem-loop may slide across the rim, as suggested by some structures of Hfq complexes. A short description of these additional results was added to page 12.

3. The experiments in Figure 5 indicating that Hfq can enable an sRNA to switch between two target sites positioned downstream of an ARN motif, employed a construct with two identical target sites. It was somewhat surprising that the target site closest the ARN motif seemed to be targeted less frequently by the sRNA than the one positioned further away because this site might be expected to be encountered first in a 1D search. The authors suggest the preferential filling of the more distal site is explained by formation of a more stable complex when this site is filled, perhaps by wrapping of the RNA with the arginine patches. However, can the authors rule out the possibility that the observed occupancy preferences aren’t due to sequences flanking the target sites that might facilitate greater pairing with the sRNA in one case when compared to the other?

The reviewer rightly points out that flanking sequences can influence pairing preferences. The work of Papenfort et al.² demonstrates that the target's seed region is typically flanked by 3' adenosine. In contrast, our tandem mRNAs feature uracil flanking the target adjacent to the ARN motif and guanosine flanking the distant target. Since neither of these flanking sequences includes adenosine, we anticipate a minimal impact on site occupancy. We added this explanation on Page 10.

Reviewer #2:

Małecka et. al elucidate the molecular mechanism of Hfq-sRNA complex dynamics with an associated mRNA molecule. The authors use single-molecule FRET to show that Hfq-sRNA can locate sRNA binding site several nucleotides away from Hfq binding site on an mRNA strand. They demonstrate that Hfq compacts the RNA strand to access the sRNA binding site and the rim interactions are necessary for the compaction. They also show sequence dependency of this compaction in that the presence of uridines between the two binding sites are essential to effectively compact the RNA.

Further investigation by the authors shows that blocking the region between the (AAC)_n and the

sRNA binding site prevents Hfq-sRNA in accessing the sRNA binding site, demonstrating the importance of the flexible segments between the two sites. Using a similar strategy, the authors also show the importance of accessibility of (AAC) site and Hfq acts as a bridge between the two binding sites. In the final set of experiments, the authors use an mRNA with tandem target sites separated by spacers to observe if Hfq shuttles between the two sites. They notice that the complex shuttles between the two target sites with the lifetimes of each target site binding shorter than that of a single binding site mRNA. It is also noticed that the shuttling probability is reduced once stable base pairing is ensued. Lastly, the authors show that the rim of the protein not only help in compaction but also help in the search of sRNA binding sites.

Based on these experiments, the authors propose a stepwise model in which Hfq-sRNA initially binds the ARN motif on the mRNA. Subsequently, the arginine residues in the rim of the protein facilitate in scanning for the complementary region to the sRNA. The scanning happens without dissociation of the protein complex and once the binding site is found, it leads to a stable duplex of sRNA-mRNA.

Overall, the experiments are well-designed, extensive, and well-thought-out. The authors have systematically endeavored to elucidate the mechanism of target finding and recognition of Hfq. This manuscript stands out as one of the most exemplary Hfq papers published recently, presenting a compelling model. It is a solid paper deserving publication in Nature Communications with only minor corrections required. I have only a few minor questions that I would appreciate the authors addressing.

Major comments

None

Minor comments

1. Fig S1B (WT Hfq) shows a small percentage of low-FRET peak which is absent/not pronounced in the other histograms. Is it donor-only peak or some other artefact?

Donor-only events have been excluded from the analysis. Nevertheless, for this mRNA, we detected a small fraction of short-lived binding events with low FRET efficiency. It is possible that a limited distance between Hfq and the sRNA binding site hindered annealing or destabilized the sRNA-mRNA duplex (Fig. 1D). However, these occurrences were infrequent.

2. In line 129-130, it is mentioned that a spacer helps recruit Hfq. Could the authors elaborate on this? Is this a consequence of using a particular spacer sequence or is it a general statement?

Please see our response to Question 2 (Reviewer 1) and explanation on Page 5 of the manuscript.

3. In Fig. S3A, the bands are denoted with “•Hfq” - does it mean that Hfq was added, or is it complexed?

RNA or RNA-tether duplexes were incubated with Hfq prior to being loaded onto a gel, leading to the formation of complexes with Hfq. We added a sentence to the figure legend for clarity.

4. In line 195, the authors explain how mRNA flexibility is essential for Hfq target search. Please elaborate how the flexibility is different from the accessibility of the mRNA. Can their method decouple the two different contributions?

YES, these two concepts are different – flexibility refers to bending of the linker RNA so that the distant site can approach Hfq bound to the ARN motif, whereas accessibility means that the site is open to Hfq and sRNA binding but without any presumption of how the site can be reached. The best evidence that flexibility is important is that annealing with the distant site is reduced if the intervening spacer is stiffened by hybridization with a complementary oligonucleotide (Fig. 4A-E). A sentence explaining why flexibility is important was added to the main text on pg. 7.

The next set of experiments in the main text mentions accessibility of the ARN motif, which has to do with whether Hfq remains bound to this motif while scanning the adjacent mRNA.

5. In the same section, (AAC)₆ motif is used instead of (AAC)₄. Wouldn't that increase the binding lifetime of Hfq and therefore make the ASO increasingly inaccessible to the (AAC) site? In other words, having (AAC)₄ would make the comparison with other experiments fairer, unless the authors show/explain that it does not make much of a difference.

4 x AAC and 6 x AAC are both strong binding sites for Hfq. We utilized the (AAC)₆ motif in experiments involving an internal label due to the availability of this mRNA fragment in the laboratory. However, we agree with the reviewer's suggestion that a more robust distal site binding motif could impact the overall efficiency of sRNA annealing. Therefore, we measured the annealing efficiency of mRNA (AAC)₆-N35-S6 before committing to experiments using the internally Cy5-labeled version of this mRNA (Fig. S4E). As expected, the sRNA successfully anneals at the anticipated location in the 6 x AAC mRNA, as evidenced by the high FRET population (Fig. S4E, middle row). To address the reviewer's question in more detail, we analyzed the lifetimes of sRNA-mRNA complexes and compared them to the lifetimes of complexes with (AAC)₄-N35-S6 mRNA. The distributions were similar, with a slightly lower mean lifetime observed for the mRNA with the (AAC)₆ motif (156 s vs. 132 s). This comparison is now shown in a new panel in the revised Fig. S4E. The difference in the lifetimes may stem from the lower number of events for the (AAC)₆ mRNA and is likely insignificant.

6. Line 280 - The average lifetime(s) of sRNA annealing events for the tandem targets are quite drastic. Is there a particular reason why one of the targets (the one farther from (AAC) site) shows more than twice the lifetime of the other one? Does it have to do with the instability of potential short loops close to the (AAC) site? If so, the frequency of sRNA transfer (Line 299) depends not only the base-pairing but also the 'flexibility' of the mRNA.

The reviewer is correct that sRNA base pairing to the distant site appears more stable, as judged from the dwell time in the high FRET state that is related to the frequency of transitions from the high FRET state to the low FRET state. We do not have a definitive explanation for this observation, but it is possible that base pairing with the site immediately adjacent to the ARN motif introduces some strain in the complex. We have noted this possibility in the text on pg. 10.

Base pairing with the distant site is at least as favorable when N = 35 as when N = 13 (Fig. 5H). However, the entropic penalty of closing a 35 nt loop is greater than the penalty for closing a 13 nt loop, if the spacer RNA is totally unstructured. From this, we infer that there are likely some additional interactions that stabilize the compact structure of the 35 nt spacer on Hfq. Our favored explanation is that some secondary structure forms in the spacer when its ends are brought together by Hfq. However, the longer spacer could also interact with the protein. At the present time, we cannot distinguish these possibilities, but either would be compatible with our final model. The potential for additional interactions is now noted on pg. 11.

Reviewer #3:

In this study, Malecka and Woodson used single molecule FRET with purified components in vitro to examine how the RNA chaperone protein assists in scanning an mRNA for a site of small RNA base pairing. The authors document that Hfq promotes compaction of the mRNA and also allows the sRNA to test binding to two different sites while remaining bound to the mRNA. These Hfq activities require the arginine residues on the rim of the Hfq hexamer.

The study is very carefully performed, and the manuscript is clearly written. The findings help to clarify how the Hfq protein can facilitate the rapid sRNA-mediated regulation observed in vivo. I only have editorial comments:

1. The following sentences were not optimally clear.

--Page 2, lines 52-53: Clarify that the scanning mechanism is being suggested for RNase E.

Corrected, with an additional reference to scanning by Cas9-sgRNA.

--Page 4, line 101: The authors should be clear that when they are always referring to the Hfq hexamer (somewhere in the text).

Done. We also added a sentence in Materials and Methods.

--Page 5, lines 129-130: The sentence "More events were observed for the longer mRNAs" seems disconnected from the sentences before and after.

Done.

--Page 6: line 155-157: I thought more explanation could be provided for the sentence beginning "These low FRET transitions..."

The text was edited and is hopefully clearer now.

--Page 8, line 206: "cannot fold the double-stranded spacer" was slightly cryptic.

Done.

--Page 10, lines 272-273: Why is annealing to the distant site preferred?

Please see the response to point 6 by reviewer 2.

--Page 10, line 274: Possibly replace "sites" with "states".

Done.

--Page 10, lines 282-284: Sentence beginning "This observation suggests that there are unresolved ..." was slightly cryptic.

This sentence was rewritten to better express our reasoning that all of the tested duplexes must be unzipped by Hfq, because they have the same sequence and the same stability (neglecting contextual considerations noted by reviewer 1). Because complexes with the single-site mRNA appear more stable than complexes with the tandem mRNA, we infer that the single site mRNA has been unzipped but these transient events go undetected in our TIRF experiment. We only see the outcome of scanning by trapping the sRNA at a second target site.

2. The authors need to double check their citations (some are numbers, some list the authors).

Done.

1. Panja, S. & Woodson, S. A. Hfq proximity and orientation controls RNA annealing. doi:10.1093/nar/gks618.

2. Papenfort, K., Bouvier, M., Mika, F., Sharma, C. M. & Vogel, J. Evidence for an autonomous 5' target recognition domain in an Hfq-associated small RNA. *Proc Natl Acad Sci U S A* **107**, 20435–20440 (2010).

REVIEWERS' COMMENTS

Reviewer #1 (Remarks to the Author):

All of my concerns have been fully addressed in the revision.

Reviewer #2 (Remarks to the Author):

The authors have satisfactorily responded to every concern raised by the referee. This referee strongly recommends the publication of the manuscript. My only minor suggestion is to modify the color of the red used in the rebuttal, as it poses a challenge for individuals with red-color blindness to distinguish between responses (red) and comments (black).